# Wideband DOA Estimation Utilizing a Hierarchical Prior Based on Variational Bayesian Inference

**Ninghui Li [1,\*], Xiaokuan Zhang [2], Binfeng Zong [2], Fan Lv [1], Jiahua Xu [1] and Zhaolong Wang [1]**

[1] Graduate School, Air Force Engineering University, Xi'an 710051, China; lf995407681@163.com (F.L.); Jiaxin_xuyan@163.com (J.X.); wangzhaolong2022@163.com (Z.W.)
[2] Air Defence and Anti-Missile School, Air Force Engineering University, Xi'an 710051, China; ezxk@sina.com (X.Z.); zongbinfeng@163.com (B.Z.)
[\*] Correspondence: li6667778882022@163.com

**Abstract:** The direction-of-arrival (DOA) estimation of wideband signals, based on sparse signal reconstruction, has recently been proposed, owing to its unique high-resolution performance. As a typical tool of sparse signal reconstruction, sparse Bayesian learning (SBL) enhances little sparsity in most works, leading to a non-robust local fitting. To significantly enhance sparsity, we proposed a novel hierarchical Bayesian prior framework, and deduced a novel iterative approach. It was discovered that the iterative approach had a lower computational complexity than the majority of current state-of-the-art algorithms. Besides, the proposed approach achieves a high angular estimation accuracy and sparsity performance, by utilizing the joint sparsity of the multiple measurement vector (MMV) models. Moreover, the approach stabilizes the estimated values between different frequencies or snapshots, so as to obtain a flat spatial spectrum. Extensive simulation results are presented, to demonstrate the superior performance of our method.

**Keywords:** direction-of-arrival; sparse Bayesian learning; hierarchical Bayesian prior; sparse recovery





## 1. Introduction

As a popular research focus in array-signal processing, DOA estimation has been developed rapidly, and applied widely in military and civilian fields, such as radar, sonar, wireless detection, mobile communication, biomedicine, and so on [1]. Substantially, the majority of the current research is based on narrowband signals, while the study for wideband-signal DOA estimation plays an important role, as the common assumption that signals coming from different directions occupy the same frequency band does not always hold up in practice [2,3]. Considering the performance of various algorithms, wideband-signal DOA estimation shows significant advantages in its resolution, estimation accuracy, and robustness to correlated sources.

Traditional DOA estimation methods for wideband signals can be roughly divided into two categories: the incoherent signal subspace methods (ISSMs) [4,5], and the coherent signal subspace methods (CSSMs) [6–8]. The common point of ISSMs and CSSMs is converting no-spectrum-aliasing wideband signals to narrowband signals for subsequent processing, while their most obvious difference is that CSSMs can handle coherent sources with a lower operational complexity than ISSMs. It is worth noting that there are two methods based on orthogonal testing: one is the test of orthogonality of projected subspaces (TOPS), and the other is the test of orthogonality of frequency subspaces (TOFS). Because the TOPS and TOFS operate within different principles, their approaches differ in DOA estimation. They all share the ability to determine the ideal reference frequency point, and strike a balance between the computational complexity and signal-to noise-ratio (SNR). However, all the above algorithms have to face the following basic limitations: (1) pre-estimation procedures are necessary for accurate angular estimation, (2) a high level of snapshot accumulation

is intractable and inevitable, (3) a priori, the number of sources is required, and (4) their performance is constrained by the SNR.

Numerous approaches have been gradually developed to manage the aforementioned limitations owing to compressed sensing and sparse recovery [9,10]. Compared with conventional approaches, they demonstrate better performance in terms of locating coherent sources, depending less on a high SNR, and operating with higher efficiency. According to the current research, the approaches relying on sparse representation can be roughly classified into three categories. The first category is based on basis pursuit, e.g., matching pursuit (MP) [11] and orthogonal matching pursuit (OMP) [12]; the second category is based on convex optimization [13–16]; and the final category is based on SBL [17,18]. It has been proven that SBL outperforms both the basis pursuit and convex optimization in sparse recovery, leading to a more accurate angular estimation [19]. However, most SBL-like methods adopt the single measurement vector (SMV) model [18] or the generalized SMV model [17], resulting in a straightforward decoupling between various snapshots. In fact, a reasonable use of temporal correlations could improve the estimation accuracy. Additionally, it has been mentioned that the MMV models are superior to the SMV models in terms of sparse performance, e.g., stronger joint sparsity and better sparse recovery [20,21].

In light of this, a novel method, based on the block-sparse Bayesian model for wideband-signal DOA estimation, is presented in this paper. The approach converts the MMV model into the block-sparse model, in order to fully retain the advantages of the MMV model and sufficiently utilize temporal correlations. As a counterpart to SBL-like methods in the wideband field, or the MMV model, our proposed approach enables the simultaneous utilization of the joint sparsity between snapshots and frequencies using tactful algorithm design. Moreover, compared with the conventional Gaussian prior, the employed prior with hierarchical structure can significantly improve the sparsity-inducing performance of the SBL.

The rest of this work is organized as follows. In Section 2, the signal model is introduced. In Section 3, the likelihood and prior are presented for variational Bayesian inference. Further, the computational complexity of various approaches is compared. The performance of our algorithm is evaluated in Section 4, and the conclusions are drawn in Section 5. For clarity, the notations used in this paper are given in Table 1.

**Table 1.** List of notations.

| Symbol | Description |
| --- | --- |
| $\mathcal{N}(\boldsymbol{\mu}, \boldsymbol{\Sigma})$ | Real Gaussian distribution with mean $\boldsymbol{\mu}$ and covariance $\boldsymbol{\Sigma}$ |
| $\mathcal{CN}(\boldsymbol{\mu}, \boldsymbol{\Sigma})$ | Complex Gaussian distribution with mean $\boldsymbol{\mu}$ and covariance $\boldsymbol{\Sigma}$ |
| $\otimes$ | Kronecker product |
| $\odot$ | Hadamard product |
| $\circ$ | Khatri–Rao product |
| $\boldsymbol{I}_N$ | $N \times N$ identity matrix |
| $const$ | Constant |
| i.i.d | Independent and identically distributed |
| $p(a\|b)$ | Conditional probability density distribution of variable $a$ with respect to variable $b$ |
| $p(a;b)$ | Probability density distribution of variable $a$ with respect to variable $b$ |
| $q(\cdot)$ | Probability density distribution |
| $\langle\cdot\rangle_{q(\cdot)}$ | Expectation with respect to $q(\cdot)$ |
| $diag(\cdot)$ | Transforming matrix into vector diagonally or transforming vector into matrix diagonally |
| $\|\cdot\|_{p,q}$ | Obtain the $l_q$ norm after finding the $l_p$ norm for each row of a matrix |

## 2. Signal Model

Assume $K$ uncorrelated far-field wideband sources, whose DOA set is $\theta = \{\theta_1, \ldots, \theta_K\}$, are received by a linear array with $N$ sensors. Without loss of generality, the received data can be modeled as:

$$\mathbf{x}(t) = \mathbf{A}\mathbf{s}(t) + \mathbf{n}(t) \tag{1}$$

where $\mathbf{A} = [\mathbf{a}(\theta_1), \ldots, \mathbf{a}(\theta_K)] \in \mathbb{C}^{N \times K}$ is the array manifold matrix with its $k$-th entry as $\mathbf{a}(\theta_k) = [\exp(-jv_1), \ldots \exp(-jv_N)]^T$, $v_n = 2\pi f d_n \sin(\theta_k)/c$, $n = 1, \ldots, N$, $d_n$ is the location of the $n$-th sensor relative to the reference one, $c$ is the velocity of light, $\mathbf{s}(t) = [s_1(t), \ldots, s_K(t)]^T$ and $\mathbf{n}(t) = [n_1(t), \ldots, n_N(t)]^T$ are the signal vector of $K$ sources, and the additive white Gaussian noise (AWGN) vector at time $t$, respectively. The time-domain data can be converted to the frequency domain by a filter bank or the discrete Fourier transform, expressed as:

$$\mathbf{x}(f_j) = \mathbf{A}(f_j)\mathbf{s}(f_j) + \mathbf{n}(f_j) \tag{2}$$

where $f_j$ illustrates that the wideband signals are separated into $J$ sub-bands, and the individual center frequency is $f_j$, $j = 1, \ldots, J$.

In order to guarantee the data independence of the same sub-band between different snapshots, the total observation time has to satisfy $T \gg 1/(BL)$, where $B$ is the bandwidth, and $L$ is the number of snapshots. If the condition holds, the frequency-domain model of wideband array signals could equivalently be treated as the time-domain model of narrowband array signals. After matched filter and snapshot accumulation, (2) is changed into:

$$\mathbf{X}_j = \mathbf{A}_j\mathbf{S}_j + \mathbf{N}_j \tag{3}$$

where $\mathbf{A}_j = \mathbf{A}(f_j)$, $\mathbf{S}_j \in \mathbb{C}^{K \times L}$ is the complex amplitude matrix for $L$ snapshots at the $j$-th frequency point, and $\mathbf{N}_j \in \mathbb{C}^{N \times L}$ is the noise matrix at the $j$-th frequency point.

To let the model be available for sparse-recovery methods, (3) needs to be converted to the following form by sparse representation, expressed as:

$$\mathbf{X}_j = \overline{\mathbf{A}}_j\mathbf{P}_j + \mathbf{N}_j \tag{4}$$

where $\overline{\mathbf{A}}_j = [\mathbf{a}_j(\overline{\theta}_1), \ldots, \mathbf{a}_j(\overline{\theta}_M)] \in \mathbb{C}^{N \times M}$ is the extended manifold matrix, $\overline{\theta}_m \in \left\{\overline{\theta}_{m=1}^M\right\}$ is yielded by discrete sampling, and $\mathbf{P}_j \in \mathbb{C}^{M \times L}$ is the solution matrix with every row representing a potential source.

Therefore, the challenge is to accomplish sparse recovery so as to obtain $\mathbf{P}_j$ from (4). According to [22], (4) can be rewritten as:

$$\mathbf{y}_j = \mathbf{\Phi}_j\mathbf{p}_j + \mathbf{n}_j \tag{5}$$

where $\mathbf{y}_j = vec(\mathbf{X}_j^T) \in \mathbb{C}^{NL \times 1}$, $\mathbf{\Phi}_j = \overline{\mathbf{A}}_j \otimes \mathbf{I}_L \in \mathbb{C}^{NL \times ML}$, $\mathbf{p}_j = vec(\mathbf{P}_j^T) \in \mathbb{C}^{ML \times 1}$, and $\mathbf{n}_j = vec(\mathbf{N}_j^T) \in \mathbb{C}^{NL \times 1}$. Equation (5) is a block-sparse model, since $\mathbf{p}_j$ is segmentally sparse (i.e., many segments only contain zeros). In this model, we only care about the locations of nonzero elements, rather than the concrete values, since different $\mathbf{p}_j$ theoretically indicates the same location of sources. Therefore, different $\mathbf{p}_j$ can be unified as $\mathbf{p}$. Considering all the frequency bins, we can obtain:

$$\mathbf{y} = \mathbf{\Psi}\mathbf{p} + \overline{\mathbf{n}} \tag{6}$$

where $\overline{\mathbf{y}} = [\mathbf{y}_1^T, \ldots, \mathbf{y}_J^T]^T \in \mathbb{C}^{NLJ \times 1}$, $\mathbf{\Psi} = [\mathbf{\Phi}_1^T, \ldots, \mathbf{\Phi}_J^T]^T \in \mathbb{C}^{NLJ \times ML}$, and $\overline{\mathbf{n}} = [\mathbf{n}_1^T, \ldots, \mathbf{n}_J^T]^T \in \mathbb{C}^{NLJ \times 1}$.

## 3. Proposed Approach

### 3.1. Bayesian Model

It is required, and crucial, to build an appropriate Bayesian model for SBL. In this paper, priors with the hierarchy structure are imposed on hidden variables, to further enhance sparsity. As for the observed variable **y**, the likelihood is:

$$p(\mathbf{y}|\mathbf{p};\delta) \sim \mathcal{CN}(\mathbf{\Psi p}, \delta^{-1}\mathbf{I}_{NLJ}) \tag{7}$$

Impose Gaussian distribution prior to the hidden variable **p**, such that:

$$p(\mathbf{p};\boldsymbol{\gamma}) \sim \mathcal{CN}(\mathbf{0}, \boldsymbol{\Sigma}) \tag{8}$$

where $\boldsymbol{\gamma} = [\gamma_1^{-1}, \dots, \gamma_M^{-1}]^T$, and $\boldsymbol{\Sigma} = diag(\boldsymbol{\gamma}) \otimes \mathbf{I}_L \in \mathbb{R}^{ML \times 1}$. Since inverse gamma distribution is conjugate to Gaussian distribution, the gamma distribution prior is adopted for each element of $\boldsymbol{\gamma}$, expressed as:

$$p(\boldsymbol{\gamma};a,b) = \prod_{m=1}^{M} \frac{b^a}{\Gamma(a)} \gamma_m^{a-1} e^{-b\gamma_m} \tag{9}$$

where $\Gamma(a) = \int_0^\infty x^{a-1}\exp(-x)dx$, $a$ is the shape parameter, and $b$ is the scale parameter. Similarly, assume $\delta$ obeys the gamma distribution, which yields:

$$p(\delta;c,d) = \frac{c^d}{\Gamma(c)} \delta^{c-1} e^{-d\delta} \tag{10}$$

where $c$ and $d$ are the corresponding shape and scale parameters, respectively. The directed acyclic graph for representing the Bayesian model is shown in Figure 1.

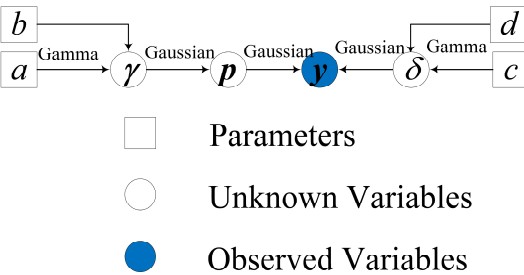

**Figure 1.** Directed acyclic graph of the proposed Bayesian model.

### 3.2. Variational Bayesian Inference

For the purpose of deriving the iterative algorithm of the SBL, Bayesian inference is adopted. Unfortunately, the closed-form solution for posterior cannot be directly obtained using Bayesian inference. However, the approximate solution can be obtained through variational Bayesian inference [22]. The posterior is factorized as:

$$p(\mathbf{p},\boldsymbol{\gamma},\delta|\mathbf{y};a,b,c,d) \approx q(\mathbf{p},\boldsymbol{\gamma},\delta) = q(\mathbf{p})q(\boldsymbol{\gamma})q(\delta) \tag{11}$$

where $q(\mathbf{p})$, $q(\boldsymbol{\gamma})$, and $q(\delta)$ are separable marginal distributions of **p**, $\boldsymbol{\gamma}$, and $\delta$. Each of their logarithmic forms can be solved by the others. As to $ln\, q(\mathbf{p})$, it satisfies

$$ln\, q(\mathbf{p}) = \langle ln\, p(\mathbf{y}|\mathbf{p},\delta)p(\mathbf{p};\boldsymbol{\gamma}) \rangle_{q(\boldsymbol{\gamma})q(\delta)} + const \tag{12}$$

In the light of (7), (8), and (12), $q(\mathbf{p})$ can be solved to obey the Gaussian distribution, with the mean and variance as:

$$\boldsymbol{\mu}_{\mathbf{p}} = \langle\delta\rangle \boldsymbol{\Sigma}_{\mathbf{p}} \boldsymbol{\Psi}^H \mathbf{y} \tag{13}$$

$$\mathbf{\Sigma_p} = \left( \langle \delta \rangle \mathbf{\Psi}^H \mathbf{\Psi} + \langle \mathbf{\Sigma} \rangle^{-1} \right)^{-1} \tag{14}$$

As usual, $NJ < M$ holds when dense sampling is applied for high-precision DOA estimation, so (13) and (14) could be equivalently transformed into the following formulae for a lower computational complexity:

$$\mathbf{\mu_p} = \langle \mathbf{\Sigma} \rangle \mathbf{\Psi}^H \left( \langle \delta \rangle^{-1} \mathbf{I}_{NLJ} + \mathbf{\Psi} \langle \mathbf{\Sigma} \rangle \mathbf{\Psi}^H \right)^{-1} \mathbf{y} \tag{15}$$

$$\mathbf{\Sigma_p} = \langle \mathbf{\Sigma} \rangle - \langle \mathbf{\Sigma} \rangle \mathbf{\Psi}^H \left( \langle \delta \rangle^{-1} \mathbf{I}_{NLJ} + \mathbf{\Psi} \langle \mathbf{\Sigma} \rangle \mathbf{\Psi}^H \right)^{-1} \mathbf{\Psi} \langle \mathbf{\Sigma} \rangle \tag{16}$$

Likewise, $q(\boldsymbol{\gamma})$ satisfies

$$ln\, q(\boldsymbol{\gamma}) = \langle ln\, p(\mathbf{p}; \boldsymbol{\gamma}) p(\boldsymbol{\gamma}; a, b) \rangle_{q(\mathbf{p})q(\delta)} + const \tag{17}$$

Utilizing (8), (9), and (17), $q(\boldsymbol{\gamma})$ is identified as a gamma distribution, whose shape parameter, $m - th$ scale parameter, and mean are:

$$\bar{a} = a + \frac{1}{2} \tag{18}$$

$$\bar{b}_m = b + \frac{1}{2} \left\langle \mathbf{p}_m^H \mathbf{p}_m \right\rangle = b + \frac{1}{2} \mathbf{\mu}_{\mathbf{p}_m}^H (\mathbf{I}_L + diag(diag(\mathbf{\Sigma}_{\mathbf{p}_m}))) \mathbf{\mu}_{\mathbf{p}_m} \tag{19}$$

$$\langle \gamma_m \rangle = \frac{\bar{a}}{\bar{b}_m} \tag{20}$$

where $\mathbf{p}_m$ and $\mathbf{\mu}_{\mathbf{p}_m} \in \mathbb{C}^{L \times 1}$ are the $m - th$ entry of $\mathbf{p} = [\mathbf{p}_1^T, \dots, \mathbf{p}_M^T]^T$ and $\mathbf{\mu_p} = [\mathbf{\mu}_{\mathbf{p}_1}^T, \dots, \mathbf{\mu}_{\mathbf{p}_M}^T]^T$, $m = 1, 2, \dots, M$, $\mathbf{\Sigma}_{\mathbf{p}_m} = \mathbf{\Sigma_p}([(m-1)L+1 : mL], [(m-1)L+1 : mL]) \in \mathbb{C}^{L \times L}$ is the sub-matrix of $\mathbf{\Sigma_p}$.

Similarly, $q(\delta)$ satisfies

$$ln\, q(\delta) = \langle ln\, p(\mathbf{y}|\mathbf{p}, \delta) p(\delta; c, d) \rangle_{q(\mathbf{p})q(\boldsymbol{\gamma})} + const \tag{21}$$

With (7), (10), and (21), $q(\delta)$ can also be solved as a gamma distribution, and its shape parameter $\bar{c}$, scale parameter $\bar{d}$, and mean are:

$$\bar{c} = c + \frac{NLJ}{2} \tag{22}$$

$$\bar{d} = d + \frac{1}{2} \left\langle (\mathbf{y} - \mathbf{\Psi} \mathbf{p})^H (\mathbf{y} - \mathbf{\Psi} \mathbf{p}) \right\rangle = d + \frac{1}{2} \mathbf{y}^H \mathbf{y} - real(\mathbf{\mu_p}^H \mathbf{\Psi}^H \mathbf{y}) + \frac{1}{2} \left\langle \mathbf{p}^H \mathbf{\Psi}^H \mathbf{\Psi} \mathbf{p} \right\rangle \tag{23}$$

$$\langle \delta \rangle = \frac{\bar{c}}{\bar{d}} \tag{24}$$

For solving the final term of (23), $\mathbf{\Psi}^H \mathbf{\Psi}$ needs to be divided into two parts; i.e., $(\mathbf{\Psi}^H \mathbf{\Psi})_{\mathbf{\Lambda}} = diag(diag(\mathbf{\Psi}^H \mathbf{\Psi}))$, and $(\mathbf{\Psi}^H \mathbf{\Psi})_{\bar{\mathbf{\Lambda}}} = \mathbf{\Psi}^H \mathbf{\Psi} - diag(diag(\mathbf{\Psi}^H \mathbf{\Psi}))$. Therefore, (23) can be rewritten as:

$$\bar{d} = d + \frac{1}{2} \mathbf{y}^H \mathbf{y} - real(\mathbf{\mu_p}^H \mathbf{\Psi}^H \mathbf{y}) + \frac{1}{2} \left\| (\mathbf{\Psi}^H \mathbf{\Psi})_{\bar{\mathbf{\Lambda}}} \odot (\mathbf{\mu_p}^H \mathbf{\mu_p}) \right\|_{1,1} + \frac{1}{2} \left\| (\mathbf{\Psi}^H \mathbf{\Psi})_{\mathbf{\Lambda}} \odot (\mathbf{\mu_p}^H \mathbf{\mu_p} + \mathbf{\Sigma_p}) \right\|_{1,1} \tag{25}$$

So far, the preparation of the proposed iterative algorithm has been finished. With the help of (15), (16), (20), and (24), it is easy to construct the iterative algorithm, in which the specific steps are as follows:

(1) **Initialization**. Set the first iterative number $k = 0$, $a = b = c = d = 10^{-6}$ (ensure uninformative distribution), and $\mathbf{p}^{(0)} = (\mathbf{\Psi}^H \mathbf{\Psi})^{-1} \mathbf{\Psi}^H \mathbf{y}$. Preset error tolerance $\varepsilon$ and maximal iterative number $k_{\max}$.

(2) **Repetition**. Input $\mathbf{p}^{(0)}$, $\varepsilon$ and $k_{\max}$.

     While $\left(\left\|(\mathbf{p}^{(k+1)} - \mathbf{p}^{(k)})/\mathbf{p}^{(k)}\right\|_2 > \varepsilon \text{ or } k < k_{\max}\right)$ do:

         {Compute $\boldsymbol{\mu_p}$ and $\boldsymbol{\Sigma_p}$ according to (15) and (16), respectively;

         Compute $\langle \boldsymbol{\Sigma} \rangle$ and $\langle \delta \rangle$ according to (20) and (24), respectively;

         Update $k = k + 1$;

         Regard $\boldsymbol{\gamma}$ as $\mathbf{p}^{(k)}$.}

     End while

(3) **Output**. Obtain final $\mathbf{p}^{(k)}$ and calculate the corresponding DOA.

### 3.3. Computational Complexity

In terms of the number of complex multiplications, as (16) dominates the computational complexity, the complexity of the proposed iterative algorithm can be expressed as $O(MN^2L^3J^2)$. When dense sampling is adopted, $O(MN^2L^3J^2)$ is less than $O(N^2M^2)$. The computational complexities of W-SpSF [15], W-SBL [16], $l_1$-SVD [14], and JLZA [13] are respectively $O(J^3M^3)$, $O(JM^3)$, $O(K^3M^3)$, and $O(M^3 + NM^2 + LNM)$. Generally, our algorithm achieves minimum computational complexity, because $M \gg N, L, J, K$ holds in most cases.

### 4. Numerical Simulation

In this section, the superior performance of our proposed approach is verified using five simulations, with ISSM [4], W-SpSF [15], W-SBL [16], $l_1$-SVD [14], and JLZA [13] compared. Root mean square error (RMSE) is used to evaluate different methods, defined as:

$$\text{RMSE} = \sqrt{\frac{1}{M_c K} \sum_{m_c=1}^{M_c} \sum_{k=1}^{K} \left(\hat{\theta}_{m_c,k} - \theta_k\right)^2} \tag{26}$$

where $M_c$ is the number of Monte Carlo trials, $\hat{\theta}_{m_c,k}$ is the estimated angle of the $k$-th source in the $m_c$-th trial, $m_c = 1, \ldots, M_c$, and $k = 1, \ldots, K$.

Here, MATLAB 2020a is used to run all the algorithms, and the platform is a ThinkStation with 512 GB RAM and 2.70 GHz CPU. If not otherwise stated, the simulation conditions are: $K = 3$ uncorrelated sources, number of sensors $N = 8$, central frequency $f_0 = 400$ Hz, bandwidth $BW = 200$ Hz, $J = 5$, $M_c = 300$, grid interval $1°$, and number of grids $M = 180$.

Simulation 1 tested the angular resolution of various algorithms. The simulation conditions were: a random DOA set $\{-5°, 35°, 60°\}$, SNR 20 dB (as similar results could be obtained at other SNR values, only one SNR value was considered here), and number of snapshots $T = 20$. As shown in Figure 2, the estimated values of the proposed approach and the ISSM were the most precise. Note that our proposed approach achieved smoother peak values, expressed low fluctuation with respect to the frequency, and showed a better joint sparsity performance than the ISSM. The results can be explained by the fact that SBL is able to converge at sparse solutions in spite of different conditions, and the proposed approach has the ability to enhance sparsity more, and stabilize the estimated values of DOA among different frequencies. To summarize, the proposed method retains the advantages of SBL, and shows a better estimation performance than ordinary SBL-like methods.

Simulation 2 examined the estimation accuracy of various algorithms. The simulation was conducted with a random DOA set $\{-5.5°, 35.1°, 60.2°\}$. In Figures 3 and 4, different $T$ and $SNR$ were adopted, to depict optimum or undesirable experimental conditions. It can be seen that the proposed approach performed the best across the whole range of SNR values and the numbers of snapshots. Among these algorithms, only the proposed approach could maintain moderate stability no matter the specific frequencies of degrees. Furthermore, the proposed approach achieved more distinct advantages when undesirable conditions were adopted. These results undoubtedly confirm that our proposed approach inherits the estimation-accuracy advantage of Bayesian compressed sensing, and further prove the superiority of the hierarchical structure in enhancing sparsity.

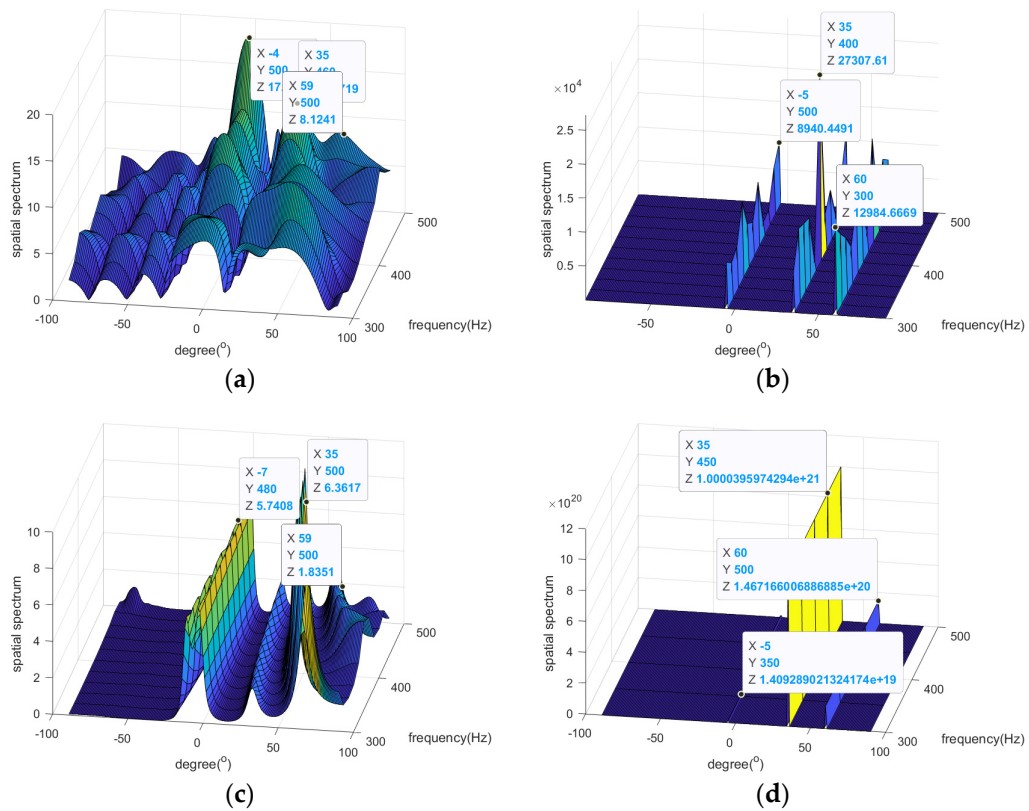

**Figure 2.** Spatial spectrum versus the degree and frequency obtained by (**a**) JLZA, (**b**) ISSM, (**c**) $l_1$-SVD, and (**d**) the proposed method.

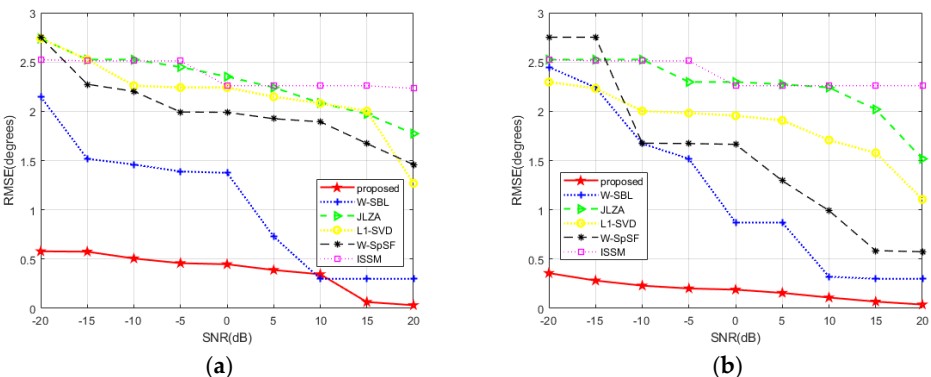

**Figure 3.** RMSE versus SNR with (**a**) $T = 5$, and (**b**) $T = 20$.

Simulation 3 tested the RMSE in relation to the sampling interval (also referred to as the grid interval). The simulation conditions were: a randomly selected DOA set $\{-30.75°, 6.47°, 44.82°\}$, SNR $20/-10$ dB, and number of snapshots $5/20$. From Figure 5, it is clear that the RMSEs of all algorithms declined when coarse grids were adopted, while the proposed method still evidently outperformed others. This indicates that coarse grids had less effect on the proposed approach than on others, as the convergence performance of SBL was robust to the grid interval. Moreover, the superiority of the proposed approach tended to strengthen even more when the SNR was low, just like in Simulation 2. Therefore, when signal-processing conditions are undesirable or inadequate, or when the sampling interval is coarse, it is preferable to adopt our proposed approach.

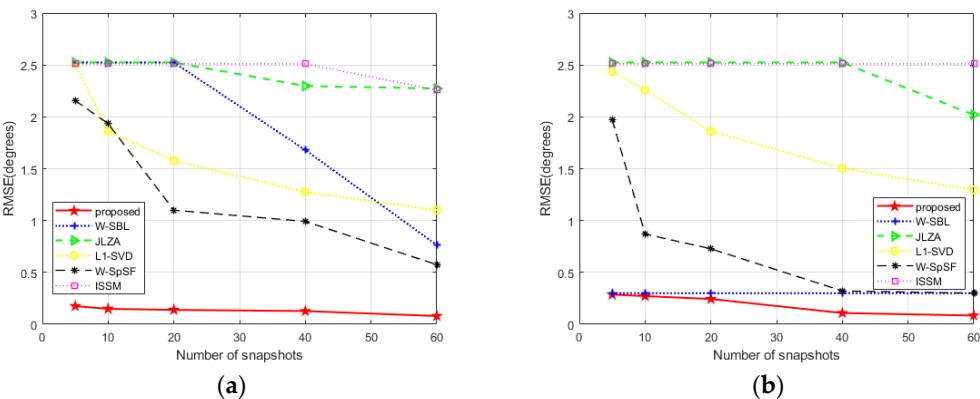

**Figure 4.** RMSE versus number of snapshots with (**a**) SNR $= -10$ dB, and (**b**) SNR $= 20$ dB.

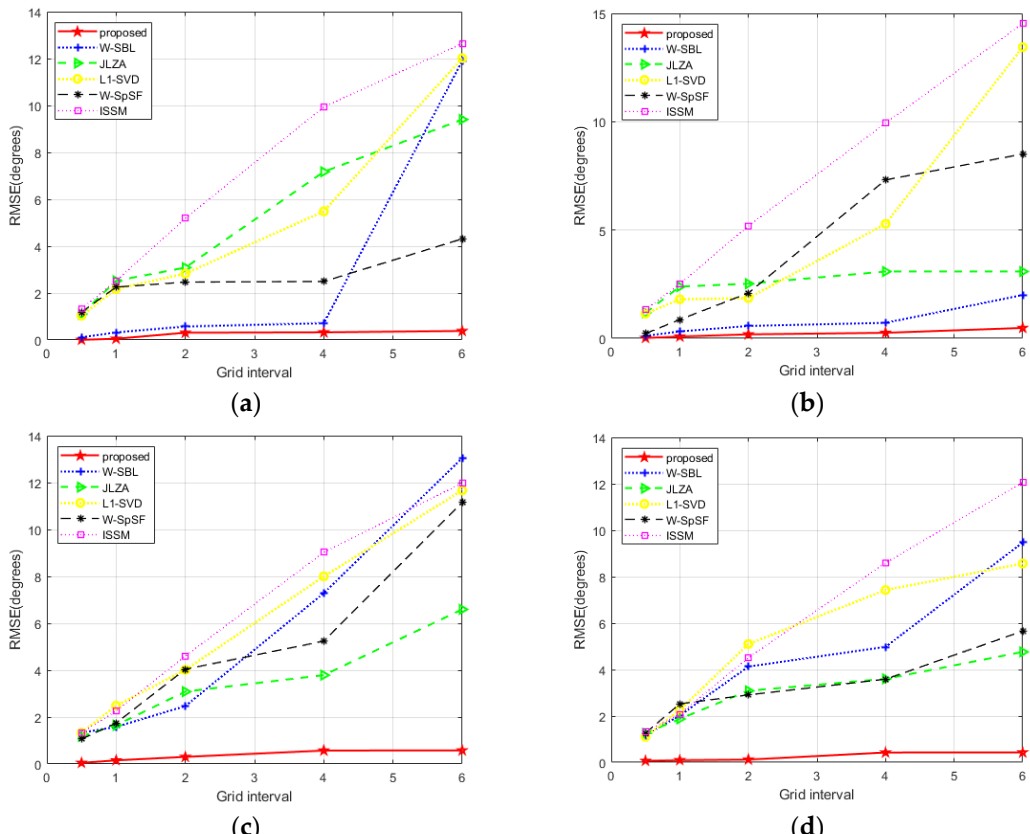

**Figure 5.** RMSE versus grid interval with (**a**) $T = 5$, SNR $= 20$ dB, (**b**) $T = 20$, SNR $= 20$ dB, (**c**) $T = 5$, SNR $= -10$ dB, and (**d**) $T = 20$, SNR $= -10$ dB.

Simulation 4 focused on the ability to detect more sources. The simulation conditions were: DOA sets with unknown sources randomly chosen from $[-90°, 90°]$, SNR $20/-10$ dB, and the number of snapshots $10/30$ (the number of snapshots was increased to allow ISSM and $l_1$-SVD to work normally). It can be seen from Figure 6 that the proposed method maintained the highest estimation precision across the whole range of numbers of sources, which confirms that the proposed approach has the potential to detect more sources.

Simulation 5 tests the dependence on the number of sensors of various algorithms. The simulation conditions were: a random DOA set $\{-30.75°, 6.47°, 44.82°\}$, SNR $20/-10$ dB, and the number of snapshots $5/20$. The results shown in Figure 7 support three conclusions: (1) the RMSE performance of the proposed approach was better than the other competitors, (2) the proposed approach could easily handle underdetermined DOA estimation problems, especially when the underdetermined degree was large, and (3) for the purpose of obtaining

accurate DOA estimated values with less antennas, the proposed approach would be a practicable selection in reality.

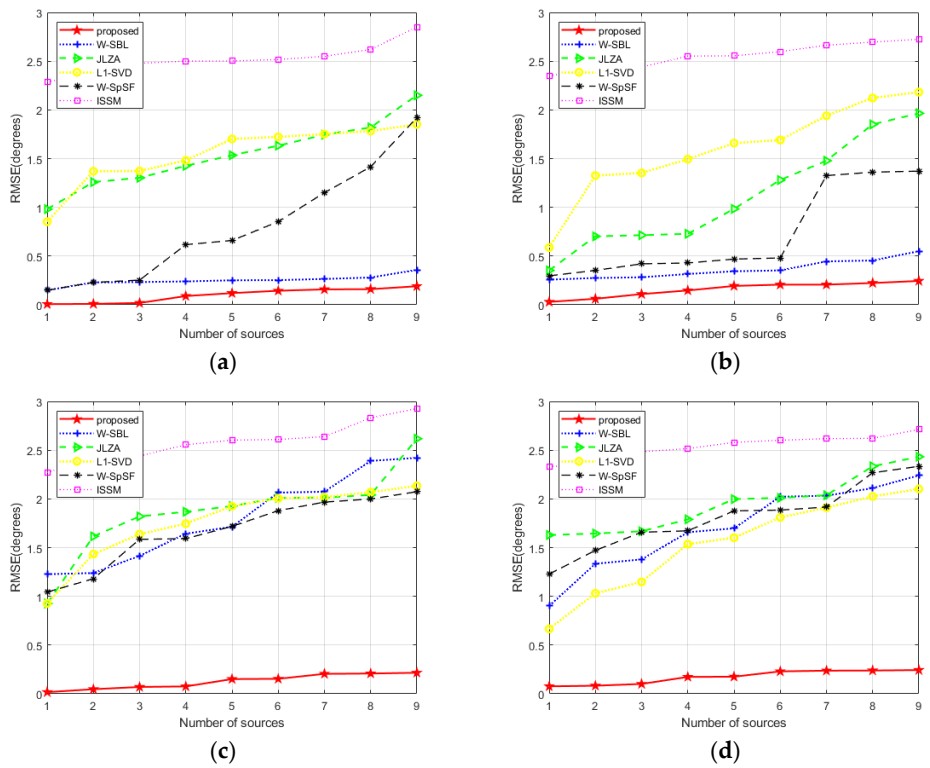

**Figure 6.** RMSE versus number of sources with (**a**) $T = 10$, SNR $= 20$ dB, (**b**) $T = 30$, SNR $= 20$ dB (**c**) $T = 10$, SNR $= -10$ dB, and (**d**) $T = 30$, SNR $= -10$ dB.

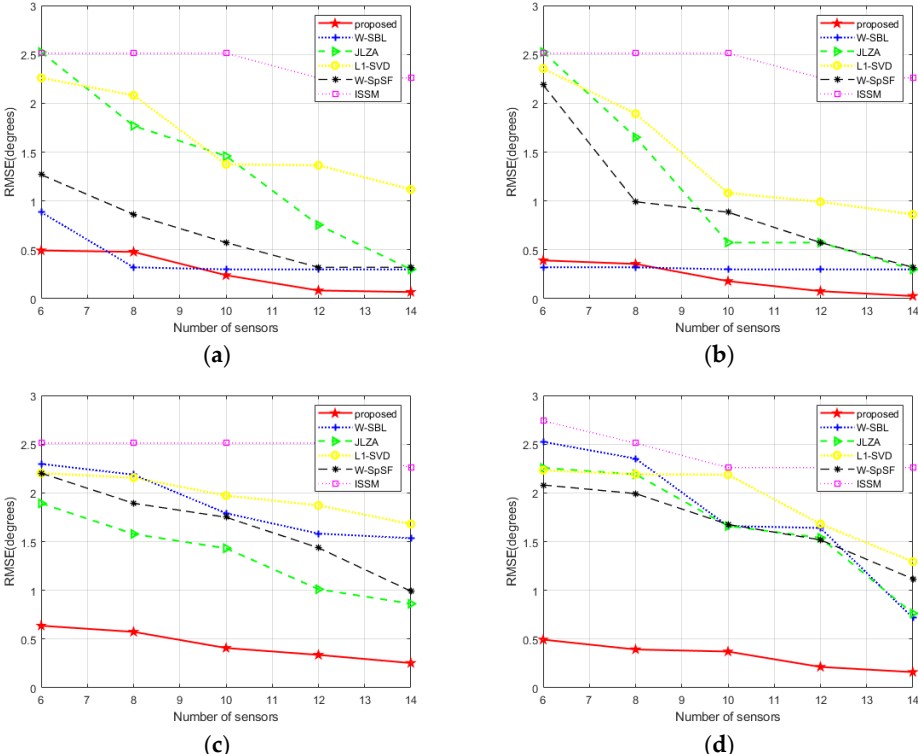

**Figure 7.** RMSE versus number of sensors with (**a**) $T = 5$, SNR $= 20$ dB, (**b**) $T = 20$, SNR $= 20$ dB, (**c**) $T = 5$, SNR $= -10$ dB, and (**d**) $T = 20$, SNR $= -10$ dB.

## 5. Conclusions

In this paper, we further extended the application of sparse Bayesian learning for DOA estimation based on wideband signals and the MMV model. The main contributions are: the hierarchical Bayesian prior framework is designed to enhance sparsity, the corresponding iterative process is derived, and the whole method is applied to solve underdetermined DOA estimation problems. In some ways, our proposed method substantially carries forward the benefits of sparse Bayesian learning, and leads to a more excellent estimation performance and sparse recovery capability. The simulation results were presented to prove the superior elements of our method, such as the sparsity enhancement, high estimation accuracy, stable estimated values, and strong adaptability.

**Author Contributions:** Methodology, N.L.; software, J.X.; validation, X.Z., B.Z.; formal analysis, F.L.; writing—original draft preparation, N.L.; writing—review and editing, Z.W. All authors have read and agreed to the published version of the manuscript.

**Funding:** This research was funded by [Natural Science Foundation of Shaanxi Province] grant number [2023-JC-YB-488].

**Data Availability Statement:** No new data were created.

**Conflicts of Interest:** The authors declare no conflict of interest.

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
