# Peer review of "Wideband DOA Estimation Utilizing a Hierarchical Prior Based on Variational Bayesian Inference"

_electronics, doi:10.3390/electronics12143074_

Round 1

Reviewer 1 Report (Previous Reviewer 1)

The paper can be published. The authors have taken into account almost all recommendation of the reviewer.

English is well.

Reviewer 2 Report (Previous Reviewer 2)

I have no further comments. The authors addressed the comments I made. I recommend to accept the manuscript in its current form.

I have no further comments. The authors addressed the comments I made. I recommend to accept the manuscript in its current form.

This manuscript is a resubmission of an earlier submission. The following is a list of the peer review reports and author responses from that submission.

Round 1

Reviewer 1 Report

The paper presented for possible publication is well written and , in my opinion, can be interesting to the readers. I recommend the authors to clarify some moments.

1) Formula (25) is unclear (may be because of very small font).

2) Clarify, please, designations in Formula (12), (17) and (21) (Inq) and also in Line  108 (Is the tensor product of two matrices meant in that Line or not?) .

3) How did You derive the computational complexity of the proposed algorithm? This issue is very important not only for applications, but  also from the theoretical point of view.

4) What  happens with the RMSE if the  key parameters of this method increase significantly? I mean K,N,M,T.

5) Give some more details of Your algorithm. What does the following phrase mean: " Compute the hyperparameters with 12, 13,17 and 22? (12) is an equation which contains unknown distribution q(p). I think, it is a function, not a parameter.

I recommend the authors to redraw Figures 3 and 4 and mark the concrete values of RMSE on these diagramms. 

The paper is well written. English Language is good.

Reviewer 2 Report

The authors presented a wideband DOA estimation technique based on variational bayesian interface. The work presented good mathematical analysis, however, this reviewer has the following comments about the work presented.

1. What kind of numerical simulation tools used to validate the proposed work? Please elaborate.

2. In experiment 1, the SNR value is chosen 20 dB. What is the basis or rationale of choosing this specific value?

3. In Figure 4, RMSE is compared with SNR value of -10 dB and 20 dB. Again, the reviewer did not find any rationale or explanation of SNR value selection. Was it randomly chosen or there is any specific reason or need to chose the values?

4. Also, for both the experiments, what are the basis of selecting the DOA sets? Again, was it randomly chosen?

5. The reviewer is concerned with validating the proposed idea with only two experiments, also the experiment parameters are not widely varied. Also, the experiments did not consider the factors like number of antennas and number of signal sources, which play a vital role in DOA estimation. 

The English language and grammar needs to be improved significantly to be consistent with standard technical writing in peer-reviewed journals.